# Emissions of Fungal Volatile Organic Compounds in Residential Environments and Temporal Emission Patterns: Implications for Sampling Methods

**DOI:** 10.3390/ijerph191912601

**Published:** 2022-10-02

**Authors:** Kyunghoon Kim, Suyeon Lee, Yelim Choi, Daekeun Kim

**Affiliations:** Department of Environmental Engineering, Seoul National University of Science and Technology, Seoul 01811, Korea

**Keywords:** mVOCs, fungi, emission, indoor materials, chamber experiment, field experiment

## Abstract

Currently, little is known about the occurrences of fungi-derived microbial volatile organic compounds (mVOCs) in various indoor materials and their detection in residential environments, despite mVOCs being linked to several acute health effects. We identified various mVOCs emitted from fungi grown on PVC wallpaper, silicone rubber, and malt extract agar. We also investigated mVOCs temporal emission and whether fungi-derived VOCs concentration can be used to estimate fungal concentration in the air using active and passive air sampling methods. Among the three fungal growth media included in this study, silicone rubber produced the most variety of mVOCs: 106 compounds (from *Aspergillus niger*), 35 compounds (from *Alternaria alternata*), and 85 compounds (from *Penicillium chrysogenum*). We also found the emission patterns of eight chemical classes (i.e., aromatics, ethers, aliphatics, alcohols, ketones, aldehydes, chlorides, and nitrides) from the three different fungi. From the results of our field experiments in 11 residential environments, passive air samplers led to higher correlations coefficients (0.08 to 0.86) between mVOCs’ air concentrations and airborne fungal concentrations, compared with active air samplers, which showed negative correlation coefficients (−0.99 to −0.02) for most compounds. This study elucidated the occurrence and temporal emission patterns of fungal VOCs in residential environments.

## 1. Introduction

Indoor exposure to volatile organic compounds (VOCs) and their potential impacts on human health have drawn public attention because people in developed countries spend more than 80% of their time in indoor environments [1], and indoor VOCs levels are several orders of magnitude higher than outdoor levels [2,3,4]. VOCs in indoor environments originate from various indoor or outdoor sources, including volatile chemical products, traffic, furniture, smoking, and fungi [5,6,7]. Human exposure to VOCs occurs mainly via inhalation, dermal uptake, and dust ingestion [8,9]. Among the three main exposure routes, inhalation is the dominant human route of exposure to VOCs [4,10]. Li et al. established a human exposure model and found that inhalation contributes to almost 100% of total human exposure to organic compounds with a logarithm of octanol–air partition coefficient less than 6.5 [10]. Furthermore, it was found that VOCs with higher indoor intake estimates also tended to have a larger proportion of intake from inhalation rather than ingestion and dermal uptake [11]. Their model results indicate that indoor air quality plays an important role in human exposure to VOCs.

Fungi, one of the indoor emission sources of VOCs, are known to deteriorate indoor air quality and negatively affect the quality of human life in multiple ways. For example, bioaerosol exposure to airborne fungi was associated with increased asthma development and allergic rhinitis in fungal-sensitive populations [12]. Fungi also produce microbial volatile organic compounds (mVOCs) in indoor environments as a result of metabolism [7], and mVOCs can enter the gas phase easily [13], resulting in human exposure via inhalation. Exposure to some mVOCs was related to several acute health effects, including headaches, eye and throat irritation, and bronchitis [14,15]. In addition, several mVOCs (e.g., 3-methyl-1-butanol, 3-octanone, 1-octen-3-ol, and formaldehyde) were also recognized as odor-causing compounds [16] and a risk factor for sick building syndrome [17].

Fungi can grow on various indoor materials, which act as fungal growth media, such as wallpaper, silicone rubber, air conditioner filters, ventilation ducts, carpets, furniture, and household dust [18,19,20,21,22], resulting in the ubiquity of fungi in indoor environments. In previous studies, profiles of mVOCs emitted from fungi are known to depend on the growth media of fungi [23,24], causing variations in the number and concentration of mVOCs with different growth media. It suggests that different mVOCs profiles can be observed depending on the types of indoor environments (e.g., library, residential environments, and hospital) as different indoor materials can be used according to the purposes of each type of indoor environment. Profiles of mVOCs can also be dependent on species of fungi. Therefore, for a thorough understanding of mVOCs indoor emissions, further studies are needed to investigate the occurrence of mVOCs emitted from species of fungi that are widely found indoors and which grow on various indoor materials.

For the determination of airborne mVOCs concentrations, different methods of sampling (i.e., active air sampling and passive air sampling) onto adsorption tubes with subsequent gas chromatography–mass spectrometry (GC–MS) have been widely used [25,26]. Active air sampling has been used for sampling mVOCs, as this method enables quick sampling. In contrast, passive air sampling has been used because of its ability to better obtain an overview regarding the airborne mVOCs concentrations, compared with active air sampling, via relatively long-term sampling over a few days or weeks [27]. Choosing an appropriate sampling method for the determination of mVOCs concentrations can help understand human fungal exposure because some mVOCs are known as indicators of fungi in indoor environments [7,28].

The objective of the current study was to investigate the various aspects of mVOCs in residential environments, including occurrences of various mVOCs emitted from fungi grown on different indoor materials, temporal emission patterns of mVOCs from fungi for a month, and the relationship between mVOCs air concentrations and airborne fungal concentrations. Specifically, we conducted chamber experiments to investigate emission characteristics of fungi-derived mVOCs from multiple fungi species (i.e., *Aspergillus niger*, *Alternaria alternata*, and *Penicillium chrysogenum*), which were detected frequently in residential environments and grown on different growth media (i.e., malt extract agar (MEA), polyvinyl chloride (PVC) wallpapers, and silicone rubber). We also conducted field experiments to find the occurrences of fungi-derived mVOCs in residential environments. Lastly, we examined the potential use of fungi-derived mVOCs concentrations as a proxy for human fungal exposure indoors by estimating correlation coefficients (r) between fungi-derived mVOCs concentrations and fungal concentrations and compared the correlation coefficients between different sampling methods (i.e., active air sampling and passive air sampling).

## 2. Materials and Methods

### 2.1. Chamber Experiments

Small-scale acrylic chambers were fabricated with a dimension of 0.3 m in length, 0.2 m in width, and 0.19 m in height (the chamber volume of 0.011 m^3^) for emission testing of mVOCs from three fungi species grown on three different growth media (Appendix A). The chambers were enclosed by polytetrafluoroethylene (PTFE) film to ensure a tight seal. The three fungi species included in this experiment were *Aspergillus niger, Alternaria alternata*, and *Penicillium chrysogenum* selected based on our previous work that detected the three species in most residential environments regardless of seasons [29]. The three selected fungi species were cultured in 10 mL of distilled water in the form of spores for more than 5 days, and then 0.5 mL of the culture solution was collected and further cultured on different fungal growth media. We used MEA (malt extract agar, Kisan Bio Co., Ltd., Seoul, Korea), PVC wallpapers, and silicone rubber (Noroo Paint & Coatings Co., Ltd., Anyang, Korea) as the fungal growth media because PVC wallpapers and silicone rubber are building interior materials that are widely used, and MEA is a material used for collecting airborne fungi indoors. The fungal growth experiments using MEA lasted for 4 weeks, and those using PVC wallpapers and silicone rubber lasted for 6 weeks, considering the potential difference in both the growth rate of fungi and the timing of mVOCs generation from fungi depending on the different growth media.

The fungal growth experiments were conducted inside the small-scale chambers with an operating temperature of 28 ± 1 °C and relative humidity of 90% maintained with K_2_SO_4_ supersaturated solution. We sampled the headspace gas from the chambers once every 3 days during the entire fungal cultivation period. For each sample collection, 11 L of headspace gas inside the chambers was collected using 99.9999% pure air with a flow rate of 0.10 L/min and introduced into an adsorbent tube filled with Tenax-TA (20913-U, Supelco, Bellefonte, PA., USA) as a sorbing material.

### 2.2. Field Experiments

To check if mVOCs detected in our chamber experiments are also detected in real residential environments, we recruited 11 households in Seoul, Korea, to obtain air samples. Specifically, we collected air samples for measuring mVOCs concentrations using two different sampling methods: active air sampling and passive air sampling. The field experiments were conducted in summer (July–August) and winter (January–February) in the same households. The characteristics of the 11 households involved in this study are available in Supplementary Material (Appendix A).

For the active air sampling, we used the Tenax-TA as a sorbent tube packed with 200 mg of 2,6-diphenylene oxide to collect the air samples. Before collecting the air samples, the tube was preconditioned using a tube conditioner (APK-1200, KNR Co., Ltd., Namyangju, Korea) for 180 min under a flow rate of 100 mL/min of pure nitrogen gas (99.9999%, Air Korea Gas Co., Ltd., Yeoju, Korea). We used an air pump (Libra Plus LP-1, A.P. Buck. Inc., Orlando, FL, USA), connected with the sorbent tube, to collect the air samples under a pump inlet flow rate of 100 mL/min for 120 min, leading to 12 L of sample volume. After collecting the air samples, we brought them to our laboratory and placed them in a 5 °C refrigerator until chemical analysis. To avoid the deterioration of the samples, we conducted the chemical analysis within 3 weeks.

For the passive air sampling, we used a passive sampler (OVM-3500, 3M Co., Ltd., Minneapolis, MN, USA) to collect the air samples. The passive samplers were placed in the households for 4 weeks and then brought to our laboratory for chemical analysis. To extract the sorbents, we injected 1 mL of CS_2_ (Kanto Chemical Co., Inc., Tokyo, Japan) inside the samplers, stirred the samplers for 30 min under 30 °C, and placed them in a −5 °C freezer until chemical analysis. We conducted the chemical analysis 3 weeks after collecting the samples from the households. The room temperature and humidity were recorded during the same time of air sampling, which was presented in the preliminary study [29]. To check if mVOCs concentrations can be used as a surrogate for fungal exposure assessment, we collected air samples from the same 11 households to measure the airborne fungal concentrations. The method for estimating fungal concentrations is available in Appendix A. For collecting bioaerosols in air samples, we used a sampling pump (Buck Bio-CultureTM, B30120, A.P. Buck Inc., Orlando, FL, USA) for 5 min per sampling cycle with a flow rate of 28.3 L/min at a sampling height of 1.2–1.5 m. We repeated this three times (three sampling cycles), and each sampling cycle was conducted after a 20 min interval. When collecting bioaerosols, we used MEA as a growth medium for fungi, which was prepared with the following composition: maltose 12.75 g/L, dextrin 2.75 g/L, peptone 0.78 g/L, and agar 15 g/L.

### 2.3. Chemical Analysis of mVOCs (Chamber Experiments)

After collecting samples from our chamber experiments, we conducted the chemical analysis of the collected samples using a thermal desorption unit (UNITY, Markes International, Bridgend, UK) coupled with gas chromatography (7820A, Agilent Technology, Santa Clara, CA, USA)–mass spectrometry (5977E, Agilent Technology, Santa Clara, CA, USA) (TD–GC–MS). Analysis conditions of TD–GC–MS are described in Appendix A. Using the Standard Reference Database (NT Search 2.0, National Institute of Standard and Technology, Gaithersburg, MD, USA), we confirmed compounds that were detected in either the headspace gas of the small-scale chambers or indoor air in residential environments (11 households).

Then, we also quantified the concentrations of the detected compounds using liquid standards (1000 μg/mL, 48 Component Indoor Air Standard, Supelco, Bellefonte, PA, USA). Calibration curves were made with the liquid standard adsorbed on the Tenax-TA tube with four different concentrations (i.e., 5, 10, 50, and 500 ng/tube). Correlation coefficients in the calibration curves were higher than 0.999 with all four concentrations, and method detection limits ranged from 0.02 μg/m^3^ (Xylene) to 0.25 μg/m^3^ (Ethylbenzene).

For field experiments, the methods of chemical analysis are described in Appendix A.

### 2.4. Statistical Analyses

To characterize the emission pattern of fungi-derived mVOCs during the cultivation period (4–6 weeks), we conducted a clustering analysis for mVOCs concentrations with different fungi growth media. The clustering analysis is a method widely used for the classification of objects into groups (also known as clusters) with a similarity in a certain factor. Specifically, in this study, we used a hierarchical clustering analysis method using TIBCO Spotfire (TIBCO Software Inc., Palo Alto, CA, USA). We also categorized the detected compounds into eight groups according to their composition (i.e., aromatics, ethers, aliphatics, alcohols, ketones, aldehydes, chlorides, and nitrides) and observed the emission patterns of each group.

We also examined the correlations between airborne fungal concentrations and mVOCs concentrations measured in 11 residential environments. To examine the appropriate sampling methods for indoor mVOCs, we examined the correlations using two different mVOCs concentrations obtained by either an active air sampler or a passive air sampler and then compared the correlations between the two sampling methods.

## 3. Results

### 3.1. Detection of mVOCs from Fungi in the Chamber Experiments

Figure 1 and Appendix A presented the number of mVOCs detected from the headspace gas of the chambers with each fungus grown on a different medium and their mass emitted during the culture period. Regardless of fungi species, silicone rubber produced the highest number of detected compounds among three growth media: 106 compounds (*A. niger*), 35 compounds (*A. alternata*), and 85 compounds (*P. chrysogenum*) (Figure 1). For all the three fungi species, the compound with the highest total emission during the whole culture period was also observed in silicone rubber; 2-ethyl-1-hexanol for *A. niger* (236 μg/c.p.), methyl methacrylate for *A. alternata* and *P. chrysogenum* (14 μg/c.p. and 220 μg/c.p., respectively).

From the viewpoint of fungal species, the highest number of detected compounds was observed in the headspace gas of the chambers cultivated with *A. niger* (108 compounds), followed by *P. chrysogenum* and *A. alternata* (88 and 43 compounds, respectively) (Appendix A). Six compounds were detected from all three growth media for *A. niger*, two compounds for *A. alternata*, and nine compounds for *P. chrysogenum* (Table 1). Most of these compounds that were detected from all three growth media also had relatively high detection frequencies—more than 50%—on each growth medium. Ethylene dichloride and methyl methacrylate were compounds emitted from all three fungal species on all three growth media.

### 3.2. Emission Pattern of Individual Classes

When sorting the compounds having detection frequencies higher than 50% into each chemical class, we observed that the emission patterns of each chemical class were similar within the same growth media across three different fungi (Figure 2). For all three fungi, emission of chloride-containing compounds (chlorides) and aromatic compounds were prevalent with MEA as a growth medium. Meanwhile, aldehydes, ketones, and alcohols were not emitted with MEA. For *A. niger* and *A. alternata*, emission patterns were observed to be similar when using PVC wallpaper as a fungal growth medium; only chlorides and ethers were emitted. In contrast, for *P. chrysogenum*, more classes of compounds (i.e., aromatics, chlorides, nitrides, and alcohols) were emitted when using PVC wallpaper as a fungal growth medium. For *A. niger* and *P. chrysogenum*, silicone rubber was observed to be the growth media that caused the widest variety of chemical classes with the highest total emissions.

### 3.3. Temporal Patterns of mVOCs Emission

Figure 3 and Appendix A present the temporal emission patterns of some of the studied mVOCs during the culture period for three individual growth media. We assigned numbers to the clusters (e.g., Cluster 1 and Cluster 2) of mVOCs when there were specific emission patterns in common among or between compounds and presented those compounds in Figure 3 and Appendix A. Depending on the combination of fungi and cultivation media, we observed 2–4 clusters having specific emission patterns.

For silicone rubber as a growth medium (Figure 3, Appendix A), we observed four clusters for *A. niger* and *P. chrysogenum* and three clusters for *A. alternata*. Compounds in Clusters 1 and 2 had relatively low concentrations throughout the cultivation period compared with Clusters 3 and 4. Compared with other growth media (i.e., MEA and wallpaper), various compounds of Clusters 3 and 4 (e.g., 2-ethyl-1-hexanol and methyl methacrylate) had consistently high natural logarithms of concentrations (>7), and their concentrations did not appear to decrease even at the end of the whole culture period. For MEA as a growth medium (Appendix A), we observed three clusters for *A. niger* and *P. chrysogenum* and two clusters for *A. alternata*. Compounds in Cluster 1 had relatively low concentrations throughout the culture period, but their emissions seemed to be continued during the same period. Overall, compounds in Cluster 2 also had relatively low concentrations for the period, but their emissions appeared to increase in the second half of the culture period (from day 18 onward) to some extent. Compounds in Cluster 3 showed relatively high concentrations, and these concentrations were observed to be consistently high; 1,1,2,2-Tetrachloroethane had the highest concentrations throughout most of the culture period with *A. niger* and *A. alternata*, while methyl methacrylate had the highest concentrations with *P. chrysogenum*.

For PVC wallpaper as a growth medium (Appendix A), we observed two clusters for *A. niger* and three clusters for *P. chrysogenum*. We did not include compounds emitted from *A. alternata* in this clustering analysis because there were only two compounds with detection frequencies higher than 50% (see Appendix A). The emission of compounds in Cluster 1 was relatively low, but it increased from the middle of the culture period. The emission of compounds in Cluster 2 occurred mostly from the beginning of the culture period to the middle. Overall, compounds in Cluster 3 had consistently high concentrations compared with those in Cluster 1 and 2, and their emissions showed an increasing tendency from the beginning of the second half. Methyl methacrylate had the highest concentrations for the whole culture period among the studied compounds.

### 3.4. Occurences of mVOCs in Residential Environments

In our preliminary test [29], physical room conditions (i.e., temperature, humidity, water damage, and ventilation) affected the fungal growth in the residential environment. Particularly, VOCs (including mVOCs) levels correlated well with indoor environmental factors when using the passive sampling method rather than the active sampling method.

Among compounds detected in more than 50% of samples in our chamber studies, we detected 13 compounds in summer and 15 compounds in winter (a total of 17 compounds) with the active air sampler (Table 2). When we used the passive air sampler, we detected 7 compounds in summer and 13 compounds in winter (a total of 13 compounds). For the most detected compounds, detection frequency was higher in winter than in summer, irrespective of sampling methods. Among the 17 compounds detected using the active air samplers, 12 compounds had higher detection frequencies in winter than in summer. Among the 13 compounds detected using the passive air samplers, 10 compounds were detected more frequently in winter than in summer.

When examining correlations between mVOCs air concentrations and airborne fungal concentrations, we found higher correlation coefficients overall with the passive air samplers (Table 2). Positive correlation coefficients (r) were observed for most compounds with the passive air samplers, ranging from 0.08 to 0.86, except for two compounds (i.e., chloroform and n-decane). In contrast, negative correlation coefficients (r) were observed for most compounds with the active air samplers, ranging from −0.99 to −0.02, except for three compounds (i.e., dimethyl disulfide, ethyl acetate, and n-butanol).

## 4. Discussion

In this study, we conducted small-scale chamber experiments to investigate occurrences of mVOCs from multiple fungi species grown on different indoor materials and investigated temporal emission patterns of mVOCs for longer than a month of cultivation. We also performed field experiments in 11 households to find occurrences of mVOCs detected in our chamber experiments and assessed the potential utility of indoor mVOCs concentrations as a proxy for human fungal exposure in residential environments. Among the three fungal growth media included in this study, we observed that silicone rubber was the growth medium that produced the highest number of detected compounds. We also found that the emission patterns of each chemical class were similar within the same growth media across three different fungi. Additionally, we found that mVOCs were classified into 2–4 clusters depending on their temporal emission patterns during the cultivation periods, and some mVOCs (e.g., methyl methacrylate and 2-ethyl-1-hexanol) were emitted at consistently high concentrations throughout the entire cultivation period, regardless of the type of cultivation media. From the results of our field experiments, we found that using passive air samplers rather than active air samplers is desirable when intending to use mVOCs concentrations as a proxy for human exposure to fungi. Lastly, we found that detection frequencies of mVOCs in residential environments in winter were higher than in summer for most compounds.

We found that mVOCs profiles were different depending on different cultivation media of fungi (Figure 1 and Appendix A), consistent with the results of previous studies [23,24]. The widest variety of mVOCs was observed when using silicone rubbers as a fungal growth medium. Silicone rubbers are one type of plastic material that can absorb some airborne organic matter and consequently facilitate the formation of biofilm (e.g., fungi and bacteria) on surfaces [30]. In addition, silicone rubbers are recognized as flexible polymeric materials that can emit a large number of organic carbon compounds over time [31,32]. The released organic carbon compounds from silicone rubbers can also enable the formation of biofilm. For these reasons, the promoted biofilm formation on silicon rubbers might contribute to the emission of various mVOCs with higher concentrations.

Our finding that fungi grown on silicone rubber produced the highest number of mVOCs may raise the issue of adverse health effects for people in residential environments. Since silicone rubber has been used in indoor residential settings in various domains such as household products, electronics, paints, and coatings [33,34], silicone rubber can act as a growth medium for fungi in various parts of indoor environments. Furthermore, from the results of our chamber experiments for temporal emission patterns of mVOCs (Figure 3, Appendix A), compared with other growth media (i.e., MEA and Wallpaper), we observed that various compounds were emitted from fungi grown on silicone rubber at consistently high concentrations. To be more specific, their emission was high (logarithm of concentration higher than 7) throughout the whole period of the cultivation experiments (~1 month) and did not tend to decrease until the end of our experiments. Considering that both fungi and fungi-derived mVOCs are known to contribute to adverse health effects [12,14,15], further studies are needed to investigate the contribution of silicone rubber to human exposure to fungi and mVOCs and develop methods to inhibit fungal growth on silicone rubber.

We examined the correlation between mVOCs concentrations and fungal concentrations and observed that correlation coefficients were higher with passive air samplers for most compounds compared with active air samplers (Table 2). Passive air samplers are known to afford long-term monitoring in field studies [27]. Considering the temporal variability of VOCs concentrations in indoor environments within a day [35], concentrations obtained from a few hours of sample collection with active air samplers are not likely to represent the long-term average concentrations. From our findings, it is recommended to use passive air samplers rather than active air samplers for collecting airborne mVOCs in future human-exposure studies. It is noted that physical room conditions, such as temperature, humidity, and ventilation, affect the sampling performance because the sampling rate in the passive air sampler is controlled by diffusion. Dodson et al. also suggested that passive air samplers are suitable for epidemiological studies involving VOCs in indoor environments [36], consistent with our suggestion. For compounds that did not have significant positive correlations between their air concentrations and fungal concentrations, they might have other primary emission sources in indoor environments besides fungi, including building materials, human activities, traffic, foodstuffs, smoking, and chemical reactions [7,37,38,39].

In this study, we also found higher detection frequencies of most mVOCs in winter than in summer (Table 2), indicating that there is a seasonal difference in the composition of mVOCs in indoor environments. Different usage patterns of consumer products that can emit VOCs indoors might contribute to the difference in the detection frequencies of mVOCs with seasons in this study [40]. Different ventilation frequencies depending on seasons might act as another factor affecting the difference in the detection frequencies with seasons [35,41]. Relatively frequent ventilation in summer might decrease the detection frequencies of mVOCs by removing mVOCs from indoor environments. In addition, the decrease in the detection frequencies of mVOCs that was due to the effect of increased ventilation is stronger for organic compounds with higher volatility [42].

We observed that mVOCs of specific chemical classes (e.g., chlorides, ethers, aliphatics, and alcohols) were prevalent in their emission from fungi compared with other chemical classes. The composition of fungal growth media might make a difference in the composition of mVOCs emitted. For example, the PVC wallpaper used in our experiment contains chloride, resulting in higher emission concentrations of chloride caused by fungal metabolism (Figure 2). In addition, silicone rubber includes oxygen in its backbone structure, which might lead to higher emission concentrations of ethers (O–R bond) or alcohols (O–H bond).

There are some limitations to this study. In addition to fungi, multiple indoor sources emit VOCs that were detected in our laboratory experiments, such as volatile chemical products, building materials, furnishings, pesticides, and combustion by-products [9,43]. Future studies are recommended to consider the multiple sources of indoor VOCs when investigating the relationship between mVOCs concentrations and airborne fungal concentrations. Moreover, recruiting more households may increase the statistical power of our correlation analysis between mVOCs concentrations and fungal concentrations. Lastly, although we have not included formaldehyde in this study, formaldehyde is one of the representative indoor environmental chemicals known to inhibit the growth of fungi. For this reason, further field studies should consider not only mVOCs levels indoors but also formaldehyde levels because formaldehyde levels can play an important role in mVOCs levels. Despite these limitations, we also noted three major strengths of this study. First, we suggest the use of passive air samplers for collecting airborne mVOCs in human fungal exposure studies rather than active air samplers. We believe that our findings can pave the way for establishing environmental policies related to sampling methods of mVOCs indoors. Second, we also found that airborne concentrations of some mVOCs can be used as a surrogate for airborne fungal concentrations in residential environments. Third, by conducting laboratory experiments and field experiments at the same time, we obtained information on various aspects of mVOCs in residential environments, including occurrences of various compounds from fungi, temporal emission patterns of mVOCs from fungi for a month, and the relationship with indoor airborne fungal concentrations.

## 5. Conclusions

The current study investigated (1) occurrences of fungi-derived mVOCs in residential environments, (2) temporal emission patterns of mVOCs from three selected fungi widely detected in residential environments, and (3) correlations between fungal concentrations and mVOCs concentrations using two different sampling methods (i.e., active air sampler and passive air sampler). We observed that a wide range of VOCs were generated from fungi in our laboratory experiment, and some compounds showed significant positive correlations between their airborne concentrations and fungal concentrations in our field experiment. Our findings may allow researchers to better understand several aspects of the indoor emission of VOCs from fungi, such as occurrences in residential environments and temporal emission patterns. Our study also suggested an appropriate sampling method (i.e., passive air sampler) to measure mVOCs concentrations in human fungal exposure studies, which will be helpful to environmental policy makers.

## Figures and Tables

**Figure 1 ijerph-19-12601-f001:**
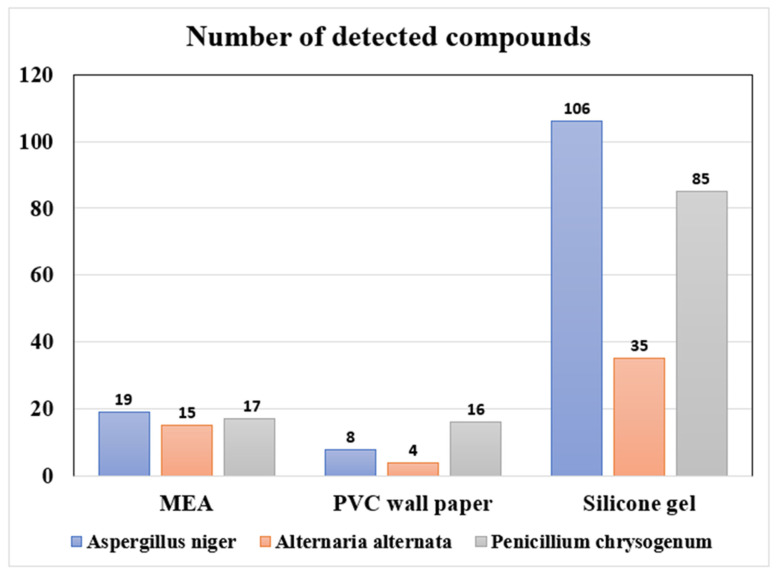
The number of compounds emitted from three fungi species cultivated on three different growth media.

**Figure 2 ijerph-19-12601-f002:**
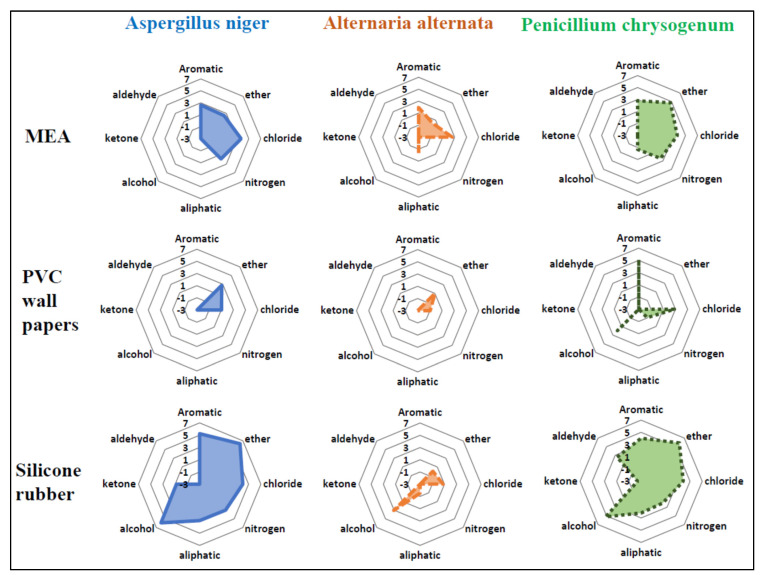
Emission patterns for eight groups of compounds with different growth media and fungi (three growth media by three fungi). The radar charts were presented on a natural logarithmic scale.

**Figure 3 ijerph-19-12601-f003:**
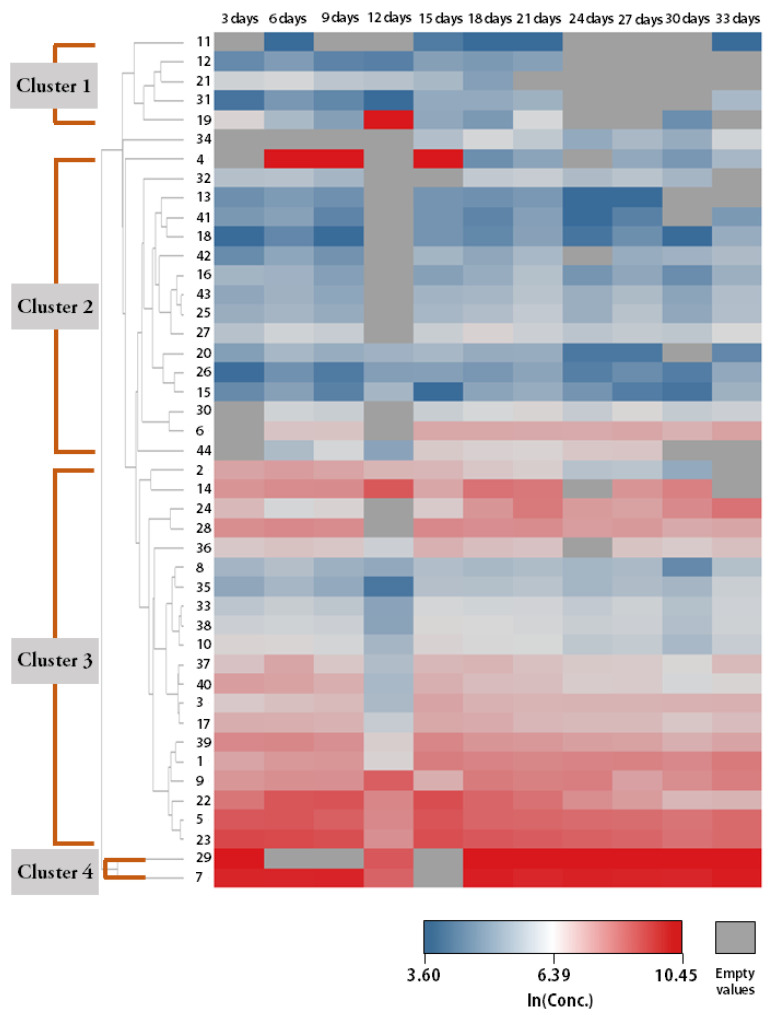
Temporal emission patterns of mVOCs originating from *Aspergillus niger* grown on silicone rubber during the cultivation period. Depending on the temporal emission patterns, compounds were classified into four clusters. Compounds were symbolized with numbers on the *y*-axis (see Appendix A for detailed information).

**Table 1 ijerph-19-12601-t001:** Compounds detected from all three growth media (i.e., MEA, PVC wallpaper, and silicon rubber) for each fungus.

Compounds	Detection Frequency (%)		
	*Aspergillus niger*	*Alternaria alternata*	*Penicillium chrysogenum*
	MEA/Wallpaper/Silicone Rubber	MEA/Wallpaper/Silicone Rubber	MEA/Wallpaper/Silicone Rubber
1,1,2,2-Tetrachloroethane	100/70/100	- *	100/100/91
Benzene	- *	- *	90/41/91
Ethyl methacrylate	40/80/100	- *	30/100/91
Ethylene dichloride	100/80/91	90/70/91	100/100/91
Isobutyronitrile	- *	- *	90/50/91
Methyl isobutyrate	50/50/91	- *	100/100/91
Methyl methacrylate	100/90/73	70/100/91	100/100/91
Methyl propionate	30/10/82	- *	- *
Methylacrylonitrile	- *	- *	90/40/82
Styrene	- *	- *	100/70/91

* Compound was not detected from all three growth media

**Table 2 ijerph-19-12601-t002:** Detection of mVOCs in 11 residential environments and correlation analysis between mVOCs air concentrations and airborne fungal concentrations.

Compounds	Active Sampling			Passive Sampling		
	DF ^a^ (Winter)	DF ^a^ (Summer)	*r* ^b^	DF ^a^ (Winter)	DF ^a^ (Summer)	*r* ^b^
2-Butanone	27	9	-	- ^c^	-	-
2-Butoxyethanol	-	9	-	-	-	-
2-Ethyl-1-hexanol	36	55	−0.04	18	27	0.08
Acetophenone	45	-	−0.81	-	-	-
Benzene	82	50	−0.35	100	82	0.56 *
Chloroform	45	-	-	-	-	−0.04
Dimethyl disulfide	55	-	0.73	9	-	-
Ethanol	-	9	-	55	36	0.23
Ethyl acetate	73	45	0.37	-	-	-
Ethylbenzene	82	81	−0.02	82	82	0.58 *
Isoprene	73	-	-	45	-	0.73
Isopropanol	-	-	-	9	-	-
n-Butanol	-	-	0.90 *	-	-	-
n-Decane	45	9	−0.32	36	-	−0.28
Nitrophenyloxadiazolol	9	18	-	54	36	0.86 *
n-Octane	89	29	-	91	-	-
o,m-Xylene	100	81	-	82	91	0.46 *
Styrene	82	44	−0.26	62	-	0.41
Toluene	82	88	−0.17	100	100	0.46 *
Trichloroethylene	-	-	−0.99	-	-	-

^a^ Detection frequency (%). ^b^ Correlation coefficient between mVOCs air concentrations and airborne fungal concentrations. ^c^ Not detected or not calculated. * *p* < 0.05. Note: Compounds were presented in this table in alphabetical order.

## Data Availability

Not applicable.

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
