# Peer review of "Emissions of Fungal Volatile Organic Compounds in Residential Environments and Temporal Emission Patterns: Implications for Sampling Methods"

_ijerph, 2022, doi:10.3390/ijerph191912601_

Round 1

Reviewer 1 Report

In the submission entitled “Emission of fungal volatile organic compounds in residential areas and temporal emission patterns: Implications for sampling methods”, Kim and the co-authors have identified and investigated temporal characteristics of emissions of various mVOCs from fungi growing on and in various indoor materials, and appropriate air sampling method has been compared between fungi concentrations and fungi-derived VOCs concentrations, consequently appropriate sampling method has been proposed. The approach is novel, and the strategy for estimating fungi concentrations is interesting and valuable. I recommend this submission be accepted for a publication in International Journal of Environmental Research and Public Health after minor revisions. My additional detailed comments are listed as follows.

1. On the title: Replace emission with emissions.

2. Line 25-26 in Abstract: Information on the relationship between mVOCs air concentrations and airborne fungal concentrations and its implication for sampling methods have not been mentioned.

3. On Introduction and Discussion: Formaldehyde is a focus in indoor air pollution. The authors should consider to better describe the formaldehyde concentration levels in Introduction section, and if possible, discuss more between mVOCs concentrations and formaldehyde concentration, so that the prospective audience could better understand the emissions and risks of mVOCs.

4. On Materials and Methods: The authors should provide more information on the deposit style and duration after production, so that chemical VOC emissions from the indoor material could be estimated by the prospective readers.

5. Line 110: Provide more significant digit number for 0.1 L/min, for example, three or at least two significant digit number should be provided.

Author Response

The comments provided by the reviewer allowed us to a significant improvement to the quality of our paper. The responses to the comments are provided below, and the manuscript has been modified accordingly. Added words have been underlined, and deleted words have been expressed with a strikethrough. The authors appreciate the comprehensive reviews.

Reviewer 2 Report

Dear authors,

Thank you for your contribution to the topic Emission of fungal volatile organic compounds in residential areas and temporal emission patterns: Implications for sampling methods. Your paper deals with an important issue of residential comfort. Although your contribution is well done, there are comments from my side.

line 25-26 and the whole manuscript

For me, a residential area means rather an area with residential buildings, https://en.wikipedia.org/wiki/Residential_area. In my opinion, there is a conflict between indoor air and residential area. Please clarify this terminology, e.g. taking into account relevant standards/guidelines, ISO 16000-29, https://www.iso.org/standard/55227.html.

You describe your sampling procedures, laboratory experiments on different substrates, their analytical results taking into account clustering. 

You have sufficiently described your laboratory experiments in terms of experimental conditions and results, 28 °C and 90 % humidity, which substances and which concentrations....

However, you do not write which conditions in indoor spaces lead to mould growth and thus to a deterioration of indoor air quality. So what parameters do fungi need to grow? Organic matter, light, water, temperature.... Therefore, elaborate a bit more on sampling and indoor conditions in the description of chapter 3.4 Occurences of mVOCs in residential areas. This connection must be clear. Otherwise it could be emitted (m)VOCs from other sources.

Please include these aspects in your introduction and also in the results of your field tests on indoor air - keyword: specification of environmental conditions. Did you apply a specific walk-through protocol? What was your personal impression of the examined rooms? According to my findings, water (humidity) and temperature, room size and window count in particular have a considerable influence on the fact that colonisation can take place at all on certain surfaces. Were you also able to determine these conditions in the examined interior rooms, especially at certain points of exterior walls (keyword heat loss and onset of condensation due to increased humidity).  This would also significantly improve your statement in lines 342-343. In lines 360-376 you discuss possible influences, but without taking into account the physical room conditions with regard to humidity and heat/temperature. What is the air mVOC-composition of the used flush medium ambient air? Have you considered also outdoor air determinations near the investigated rooms?

My overall recommendation is: Reconsinder after major revision of the results and room parameters in section 3.4.

Author Response

(The authors gave the same response as above.)

Round 2

Reviewer 2 Report

Dear Authors,

Thank you very much for your editorial update. 
There are no further comments from my side.